# Endoscopic Ultrasound (EUS) in Gastric Cancer: Current Applications and Future Perspectives

**DOI:** 10.3390/diseases13080234

**Published:** 2025-07-24

**Authors:** Dimitrios I. Ziogas, Nikolaos Kalakos, Anastasios Manolakis, Theodoros Voulgaris, Ioannis Vezakis, Mario Tadic, Ioannis S. Papanikolaou

**Affiliations:** 1Department of Gastroenterology, Athens Naval Hospital, 11521 Athens, Greece; dimiziog95@gmail.com; 2Department of Gastroenterology, General Hospital of Athens “G. Gennimatas”, 11527 Athens, Greece; nkalakos@hotmail.com; 3Department of Gastroenterology, University Hospital of Larisa, 41110 Larisa, Greece; manolakis@uth.gr; 4Academic Department of Gastroenterology, Medical School of National and Kapodistrian, University of Athens, General Hospital of Athens “Laiko”, 11527 Athens, Greece; thvoulgaris@med.uoa.gr; 5Biomedical Engineering Laboratory, School of Electrical and Computer Engineering, National Technical University of Athens, Zografou Polytechnic Campus, 15772 Athens, Greece; ivezakis@biomed.ntua.gr; 6Department of Gastroenterology, Hepatology and Clinical Nutrition, Faculty of Pharmacy and Biochemistry, University of Zagreb, Clinical Hospital Dubrava, 10040 Zagreb, Croatia; mtadic1@gmail.com; 7Hepatogastroenterology Unit, Second Department of Internal Medicine-Propaedeutic, Medical School, National and Kapodistrian University of Athens, 12462 Athens, Greece

**Keywords:** gastric cancer EUS staging, gastric cancer staging, endosonography gastric cancer, miniprobe EUS, EUS-guided gastroenterostomy

## Abstract

Gastric cancer remains the fourth leading cause of cancer-related mortality worldwide. Advanced disease is associated with a poor prognosis, emphasizing the critical importance of early diagnosis through endoscopy. In addition to prognosis, disease extent also plays a pivotal role in guiding management strategies. Therefore, accurate locoregional staging (T and N staging) is vital for optimal prognostic and therapeutic planning. Endoscopic ultrasound (EUS) has long been an essential tool in this regard, with computed tomography (CT) and, more recently, positron emission tomography–computed tomography (PET–CT) serving as alternative imaging modalities. EUS is particularly valuable in the assessment of early gastric cancer, defined as tumor invasion confined to the mucosa or submucosa. These tumors are increasingly managed by endoscopic resection techniques offering improved post-treatment quality of life. EUS has also recently been utilized in the restaging process after neoadjuvant chemotherapy, aiding in the evaluation of tumor resectability and prognosis. Its performance may be further enhanced through the application of emerging techniques such as contrast-enhanced endosonography, EUS elastography, and artificial intelligence systems. In advanced, unresectable disease, complications such as gastric outlet obstruction (GOO) severely impact patient quality of life. In this setting, EUS-guided gastroenterostomy (EUS-GE) offers a less invasive alternative to surgical gastrojejunostomy. This review summarizes and critically analyzes the role of EUS in the context of gastric cancer, highlighting its applications across different stages of the disease and evaluating its performance relative to other diagnostic modalities.

## 1. Introduction

Gastric cancer is the fourth leading cause of cancer-related mortality, with a global incidence of 11.1 per 100,000 and a mortality rate of 7.7 per 100,000 in 2020 [1,2]. The highest incidence and mortality rates are observed in Asia, followed by Latin America, the Caribbean, and Europe [1]. Gastric cancer is also twice as prevalent in males as in females [1]. Notably, while the 5-year survival rate in the United States is only 32%, largely due to delayed diagnosis from subtle early symptoms, survival rates in Eastern countries such as Korea and Japan reach 77% and 81%, respectively [3,4]. This improved prognosis is attributed to national screening programs, surveillance, and advanced therapeutic strategies [4]. Besides early diagnosis, accurate staging is also crucial, as different treatments or combinations are indicated depending on the specific disease stage. For example, with the evolution of advanced endoscopic therapies such as Endoscopic Submucosal Dissection (ESD) and Endoscopic Mucosal Resection (EMR), many patients with early-stage disease are now amenable to endoscopic therapy, offering similar curative rates as surgery [5]. In contrast, more advanced localized stages require surgical resection, with or without preoperative chemotherapy. Endoscopic ultrasound (EUS), first reported in 1980, is widely used for both diagnostic and therapeutic purposes in gastrointestinal lesions [6]. Its ability to clearly visualize the layers of the gastric wall has made it an essential tool for primary tumor (T) staging, while it has also played a role in the assessment of lymph node status (N staging). This review aims to present a holistic evaluation of the role of EUS throughout the course of gastric cancer by providing recent insights regarding its use in initial locoregional staging, restaging after neoadjuvant chemotherapy, and its potential therapeutic role in unresectable cases causing gastric outlet obstruction (GOO). Moreover, a comparison with other imaging modalities is provided, and the role of EUS relative to conventional endoscopic assessment is analyzed, in order to offer a clearer understanding of the optimal diagnostic strategy for these patients. Finally, the role of advanced EUS modalities and artificial intelligence (AI) as complementary tools to EUS is discussed.

## 2. Methods

A search of the PubMed and EMBASE database was conducted for English-language studies published up to May 2025, using the following keywords: gastric cancer staging, gastric cancer EUS staging, endosonography gastric cancer, miniprobe EUS, gastric cancer restaging, and EUS-guided gastroenterostomy. In order to improve the outcomes, we employed a stepwise strategy, conducting multiple separate searches and subsequently combining the results. Priority was placed on meta-analyses, recent clinical trials, and prospective studies to ensure the inclusion of high-quality evidence.

## 3. EUS and Gastric Cancer: Technical Considerations

EUS combines endoscopic and ultrasound imaging using an echoendoscope with an ultrasound transducer equipped at its distal tip. Two types of echoendoscopes are commonly used: radial and linear [7]. Radial EUS provides a 360-degree image and is only used for diagnostic purposes, and therefore its use is becoming limited nowadays. In contrast, linear EUS offers an oblique view and allows for interventional procedures such as fine-needle aspiration (FNA) or biopsy, making it suitable for both diagnostic and therapeutic applications [7]. To examine the gastric wall, air is removed from the stomach, and a water-filled balloon or de-aerated water is used to reduce artifacts. Most EUS probes with frequencies of 7.5–12 MHz can visualize the gastric wall as a five-layer structure with a penetration depth of 1–6 cm [8]. Normal values for gastric wall thickness remain undefined; however, a range of 2–4 mm is generally accepted [8,9]. In the five-layer pattern, layer 1 (the superficial mucosa or lumen interface) appears hyperechoic; layer 2 (muscularis mucosa or deep mucosa) is hypoechoic; layer 3 (submucosa) is hyperechoic; layer 4 (muscularis propria) is hypoechoic; and layer 5 (serosa) appears hyperechoic (Figure 1A) [9]. Higher-frequency probes (e.g., 20 MHz) yield more detailed imaging at the cost of reduced depth penetration. There are three main macroscopic types of gastric cancer, namely, protrusive (type I), superficial (type II), and excavated (type III), with the superficial type further subdivided into elevated, non-protruding, and depressed forms [8]. Histological classification includes intestinal, diffuse, and mixed types [10]. On EUS, gastric cancer is typically manifest as echo-poor, inhomogeneous thickening of the gastric wall, involving specific layers, depending on the tumor stage (Figure 1B). Occasionally, it may extend beyond the gastric wall and involve adjacent structures [9]. Linitis plastica may appear as a homogenous hypoechoic band or a preserved layered structure with thickened muscularis propria. A detailed tumor assessment by EUS is generally feasible, and a retrospective study has suggested that a threshold of 65 examinations is required for an echoendoscopist to achieve competency [11]. Nevertheless, even when adequate expertise is available, certain regions of the stomach—such as the lesser curvature (particularly at the angular fold) and the subcardial area—may remain technically challenging, potentially making tumors in these locations more difficult to visualize and stage accurately [9].

## 4. EUS in Gastric Cancer T Staging

### 4.1. General Findings

The TNM classification defines the following four categories for gastric cancer T staging: T1, characterized by invasion into the mucosa or submucosa; T2, where the tumor invades the muscularis propria; T3, with subserosal invasion; and T4, where the tumor extends to the serosa [12]. EUS has long been the cornerstone in the locoregional staging of gastric cancer, although its reliability and consistency have been increasingly called into question [13]. Cardoso et al. [14] conducted a systematic review and meta-analysis of 22 studies to evaluate the effectiveness of EUS in the T staging of gastric cancer, reporting an overall accuracy ranging from 56.9% to 87.7%. The pooled accuracies for each T stage were as follows: 77% for T1, 65% for T2, 85% for T3, and 79% for T4. In another meta-analysis, EUS demonstrated a pooled sensitivity of 88.1% and specificity of 100% for T1, 82.3% and 95.6% for T2, 89.7% and 94.7% for T3, and 99.2% and 96.7% for T4, respectively [15]. Moreover, a systematic review found an overall accuracy for T staging between 65% and 92.1%, with sensitivity in identifying serosal invasion between 77.8% and 100% [16]. Given these results, it appears that EUS demonstrates greater accuracy for advanced disease stages (T3 and T4) (Table 1) [14,15]. However, when considering cancer staging, it is paramount that a diagnostic tool effectively impacts clinical decisions regarding patient management. In fact, for tumors with muscularis invasion and beyond, the recommended approach is identical and includes a combination of neoadjuvant chemotherapy and surgical resection; thus, EUS differentiation among T2, T3, and T4 stages has limited influence on therapeutic decisions [17]. In contrast, the treatment plan differs for T1 stage tumors, referred to as early gastric cancer (EGC). These tumors can be adequately treated with either surgical or endoscopic resection without the need for preoperative chemotherapy [17]. In this context, a meta-analysis of 46 studies involving 2742 patients reported that EUS demonstrated a sensitivity of 85% (95% CI, 78–91%) and a specificity of 90% (95% CI, 85–93%) for distinguishing between T1 and T2 tumors [18]. In another meta-analysis, EUS exhibited sensitivities of 88% (95% CI, 84.5–91.1%) and 82% (95% CI, 78.2–86.0%) for identifying T1 and T2 stages, respectively [15].

### 4.2. EUS for Depth Assessment in EGC

Once the diagnosis of EGC is confirmed, it is equally crucial to determine whether the tumor is confined to the mucosa or has invaded the submucosa, as well as to estimate the depth of invasion, in order to evaluate whether the patient is a candidate for endoscopic therapy. In fact, ESD or EMR are recommended for lesions confined to the mucosa or with submucosal invasion less than 500 μm, while radical gastrectomy is indicated for deeper submucosal invasion [21]. In addition to preserving a better quality of life, endoscopic therapy is also associated with improved prognosis in patients with EGC when compared to surgical treatment [22]. Moreover, there is a considerable difference in the incidence of lymph node metastasis between mucosal and submucosal tumors, with rates of approximately 2–5% for the former and up to 30% for the latter [23,24]. This emphasizes the need for a thorough evaluation for nodal metastasis in cases of submucosal invasion, as these patients require a combined treatment approach of neoadjuvant chemotherapy and surgery. On EUS, mucosal cancers are defined as lesions confined to the first and second wall layers, whereas submucosal cancers extend into the third layer [9]. The reported accuracy of EUS in predicting the depth of invasion in EGC ranges from 56% to 88% [25,26,27,28]. Shi et al. [20] performed a meta-analysis of 17 studies, primarily retrospective, and reported a sensitivity of 87% (95% CI, 86–88%) and a specificity of 67% (95% CI, 65–70%). In addition, they found high rates of overestimation and underestimation of submucosal invasion at 33.8% and 29.7%, respectively, concluding that EUS exhibits only modest accuracy. Similarly, in another meta-analysis, EUS demonstrated a pooled sensitivity of 76% (95% CI, 74–78%) and a specificity of 72% (95% CI, 69–75%) for mucosal cancer, while the pooled sensitivity and specificity for submucosal cancer were 62% (95% CI, 59–66%) and 78% (95% CI, 76–80%), respectively [19]. Notably, a high overestimation rate of 42% for submucosal cancer was also observed. Overestimating submucosal invasion can lead to unnecessary surgery and its associated risks, while underdiagnosis may result in an incomplete cure after endoscopic therapy. This concern was highlighted in a recent retrospective study where 34% of patients who were underestimated as having mucosal cancers by EUS and treated with ESD required additional surgery [29]. To improve accuracy in estimating deep submucosal invasion and assessing the feasibility of endoscopic therapy, some studies have used a threshold of up to 1 mm, reporting promising results [30]. Other reports have challenged the necessity of EUS in the staging of EGC, as it has not proven superior to endoscopy in predicting invasion depth [31,32]. Among these, a prospective study by Choi et al. involving 955 patients found no advantage of EUS over endoscopy, with accuracies of 67.4% (644/955; 95% CI, 64.4–70.4%) and 73.7% (704/955; 95% CI, 70.9–76.5%), respectively (*p* = 0.003) [31]. To our knowledge, only one study has directly compared linear and radial echoendoscopes for depth evaluation in EGC. This was a prospective study of 72 patients conducted by Lan et al. [33], which found that linear EUS had significantly greater accuracy than radial EUS in diagnosing submucosal invasion (90.9% vs. 69.2%, *p* = 0.024). A potential advantage of linear EUS is the more feasible examination of tumors located in anatomically challenging regions of the stomach, such as the fundus and antrum, compared to radial EUS.

### 4.3. Miniprobe EUS for EGC

Miniprobe EUS represents an alternative to conventional EUS (radial and linear), utilizing a miniature ultrasound probe that can be inserted through the working channel of an endoscope and operates at high frequencies of up to 30 MHz. This enables a more detailed evaluation of the gastric wall layers and provides a precise prediction of tumor invasion, particularly in small and superficial lesions, which represent the majority of EGCs [20]. Studies that exclusively employed miniprobes have reported an overall accuracy ranging from 79% to 87% in predicting the invasion depth of EGC, with higher accuracy observed in differentiated lesions measuring < 2 cm [34,35,36]. According to the aforementioned meta-analysis by Shi et al. [20], miniprobe EUS demonstrated superior sensitivity and specificity for diagnosing the invasion depth of EGC compared to other types of EUS (90% vs. 81% and 71% vs. 63%, respectively). In another prospective study, miniprobe EUS was associated with significantly greater accuracy compared to radial echoendoscope (79.5% vs. 59.6%; *p* < 0.001) [31]. Nonetheless, other reports found comparable accuracy rates between the two methods [19,26,37]. An advantage of using miniprobes is that their detailed visualization of gastric wall layers enables the identification of well-defined patterns indicative of submucosal cancer. This is particularly important for distinguishing submucosal invasion from ulceration or fibrosis, which poses a challenge and is a significant source of misdiagnosis. That is, in their recent retrospective study, Kim et al. [22] defined an arch-shaped irregularity as deep submucosal invasion and found that this pattern was significantly associated with histologically proven invasion and predicted a lack of curative treatment after ESD (*p* < 0.001). Moreover, miniprobe EUS can be easily combined with endoscopic assessment of EGC during the same session. Recently, a combined approach using both EUS and endoscopy has been proposed for staging EGC, showing improved diagnostic accuracy compared to endoscopy alone. In particular, a multicenter prospective study evaluated 180 patients and found that miniprobe EUS was significantly effective in cases where a definitive diagnosis of submucosal invasion could not be made by endoscopy, with an accuracy of 67.1% compared to 29.3% for the latter (*p* < 0.001) [38]. Consequently, the authors suggested that EUS examination would be a reasonable approach for lesions with uncertain submucosal invasion after endoscopy.

### 4.4. Factors Influencing the Diagnostic Accuracy of EUS

In addition to the EUS type, there are a few factors that could also influence the diagnostic capability of EUS. A meta-analysis on this topic reported that the presence of ulceration, a lesion size of ≥2 cm, and an undifferentiated type of cancer significantly reduced diagnostic accuracy, with total misdiagnosis rates of 27.7%, 30.4%, and 26.9%, respectively [20]. Numerous reports have demonstrated that larger lesion size is associated with an increased risk of overestimating invasion depth, particularly in tumors > 3 cm [39,40]. Similarly, in previous studies, the presence of ulceration has been correlated with inaccurate prediction of the T stage, particularly overestimation of submucosal invasion [31,40,41]. This is because the accompanying submucosal fibrosis is often difficult to distinguish from cancer, as both appear hypoechoic on EUS assessment [31]. Regarding tumor location, most studies have revealed an increased rate of underdiagnosis for lesions in the upper third of the stomach compared to those in other gastric regions [33,42,43]. Tsuzuki et al. [42] suggested that underdiagnosis of submucosal invasion is more frequent in this region, as the submucosa is often thinner and more challenging to assess.

## 5. EUS in Gastric Cancer N Staging

The presence of nodal metastasis is of paramount importance in the management of gastric cancer. Patients with nodal involvement are considered advanced cases, requiring combined treatment with neoadjuvant chemotherapy and surgery [17]. Moreover, lymph node spread affects prognosis, as it is associated with reduced survival rates and an increased risk of recurrence [44]. Therefore, accurate N staging is essential to guiding clinical decisions. Despite the lack of standardized criteria, most experts define malignant lymph nodes on EUS as roundish, hypoechoic, well-delineated lesions, with a cut-off size ranging from 5 mm to 10 mm [9,45]. Heterogeneous meta-analyses have reported the pooled sensitivity and specificity of EUS for N staging to be approximately 83% and 67%, respectively [18,46]. It was suggested that EUS is reliable for identifying nodal metastasis, while its relatively low specificity highlights the challenges in excluding the presence of malignant lymph nodes, which often results in an overestimation of the N stage. However, in other reports, EUS demonstrated limited effectiveness in detecting nodal metastases, with sensitivity ranging from 49% to 59% [14,47,48]. Serrano et al. [49], in a retrospective study of 614 patients, reported an overall accuracy of 57% and found that the primary reason for misdiagnosis was the underestimation of the N stage, which occurred in 30% of patients. The reduced sensitivity and accuracy observed in the studies above may be explained by the fact that malignant lymph nodes are often small or of normal size, leading to their misdiagnosis during assessment [45]. That is, in a prospective study, 55% of malignant lymph nodes were smaller than 5 mm, which is a common threshold for malignancy used by many echoendoscopists [50]. EUS-guided fine needle aspiration (EUS-FNA) is a diagnostic tool that may help overcome the aforementioned challenges of conventional EUS, as it enables the sampling of suspected lymph nodes for cytological analysis. A recent meta-analysis consisting of 26 studies reported that EUS-FNA had a pooled sensitivity and specificity of 87% and 100%, respectively, for discriminating between malignant and benign lymph nodes [51]. However, this study was not focused on gastric cancer, as it included lymph nodes from various cancers. In the only study assessing the performance of EUS-FNA in N staging specifically for gastric cancer, Hassan et al. [52] found that EUS-FNA was crucial in patient evaluation, as its implementation changed the treatment plan in 15% of cases.

## 6. The Role of EUS in Gastric Cancer Restaging After Neoadjuvant Chemotherapy

Despite advancements in diagnostic techniques, most cases of gastric cancer are detected at an advanced locoregional stage (stage IB–III; >T1 and/or ≥N0M0 according to the TNM classification), requiring surgical resection. Neoadjuvant chemotherapy is indicated prior to surgery, as it is associated with improved surgical outcomes (higher R0 resection rates) and overall prognosis compared to surgery alone [53]. Consequently, a thorough restaging to assess tumor response after neoadjuvant therapy is crucial for determining resectability and prognosis. Current guidelines advocate for CT or PET–CT as the preferred modalities over EUS in this setting [17,54]. Indeed, only a few studies have evaluated the role of EUS in restaging following neoadjuvant therapy (Table 2). A recent meta-analysis of six studies reported poor performance of EUS with pooled sensitivities of 29–71% and 53% for T and N staging, respectively. EUS demonstrated particularly low sensitivity for T2 stage (29%), whereas sensitivity for T3 stage was deemed acceptable (71%). Notably, high diagnostic accuracy in differentiating between T1–T2 and T3–T4 tumor stages was also observed [55]. Similarly, a recent retrospective study of 97 patients found limited diagnostic utility of EUS, with accuracy rates of 44.4% and 49.3% for T and N restaging, respectively [56]. A prospective study of 67 patients concluded that the unsatisfactory accuracy of EUS was primarily due to a high overstaging rate, occurring in approximately 60% of cases, rather than understaging (20%) [57]. Neoadjuvant chemotherapy is associated with local inflammation and fibrosis resulting from tumor destruction. As previously mentioned, the presence of fibrosis hampers the diagnostic accuracy of EUS for T staging, which relies on a clear view of the gastric wall layers [55]. This may explain the reduced accuracy of EUS following neoadjuvant therapy. Additionally, lymph node size reduction following chemotherapy may lead EUS to falsely classify malignant nodes as non-malignant, thereby decreasing its diagnostic performance. Despite these limitations, EUS may have a prognostic value. Bohle et al. [57], in their prospective study, reported that the following features assessed by EUS were associated with a significantly favorable prognosis in terms of tumor recurrence: (a) post-chemotherapy stage T0–2 versus T3–4 (median relapse-free follow-up not reached vs. 17.7 months, *p* = 0.017); (b) downstaging of two or more stages (median relapse-free period not reached vs. 20.5 months, *p* = 0.043); and (c) post-chemotherapy tumor thickness < 15 mm versus > 15 mm (median relapse-free period not reached vs. 10.2 months, *p* = 0.003). Moreover, Hoibian et al. [56] found that among 67 patients who underwent evaluation after neoadjuvant chemotherapy, the median overall survival was 64.7 months for those who demonstrated a treatment response on EUS, compared to 22.9 months for those with stable disease (*p* = 0.01).

## 7. EUS vs. Other Diagnostic Modalities

Whether endoscopic ultrasound or computed tomography offers greater accuracy in staging gastric cancer remains a matter of debate (Table 3). Since the advent of multidetector-row computed tomography (MDCT), numerous studies have assessed its performance relative to EUS, demonstrating comparable accuracy in both T and N staging [8,13,59]. MDCT may offer greater value than EUS in cases where ulceration is present. In a retrospective study of 141 patients, both modalities showed reduced accuracy in large tumors or those located at the cardia or gastric angle; however, MDCT demonstrated significantly higher accuracy than EUS in ulcerated lesions (61.5% vs. 30.8%, *p* < 0.0001) [60]. Two heterogeneous meta-analyses have shown that EUS performs better than MDCT in T1 staging (AUC 0.903 vs. 0.774), while no significant differences have been observed for T2–T4 stages [61,62]. These findings suggest that EUS may be superior when evaluating patients with EGC and thus determining suitability for endoscopic resection, perhaps due to its ability to assess gastric wall layers more clearly than MDCT. Regarding N staging, the same meta-analyses reported that EUS is more accurate in predicting the presence or absence of lymph node involvement (N+ or N0 stage), whereas MDCT is more valuable in identifying advanced lymph node involvement, with sensitivities of 56% vs. 30% for N2 stage and 21% vs. 16% for N3 stage, respectively [60,61]. Only a handful of studies have compared the performance of EUS and PET–CT for gastric cancer staging. Redondo-Cerezo et al. [63] performed a prospective study involving 256 patients to evaluate the performance of both techniques in the preoperative N staging of gastric cancer and reported that EUS demonstrated significantly higher accuracy for both initial staging and restaging after neoadjuvant therapy (76.2% vs. 72.5%, *p* = 0.02, and 88.5% vs. 69.0%, *p* < 0.0001, respectively). Similarly, in a recent retrospective study, EUS was superior to PET–CT in N restaging (accuracy of 70.8% vs. 60.4%, respectively), and both techniques demonstrated predictive value for prognosis [64]. While EUS appears to perform better than PET–CT in restaging nodal involvement after neoadjuvant therapy, both reports emphasized that a combined approach would be reasonable, as neither modality alone provides sufficient accuracy [64].

## 8. Therapeutic Applications of EUS in Gastric Cancer

GOO refers to the blockage at the level of the antrum or duodenum, caused by either benign or malignant conditions. It is often associated with advanced, unresectable cases of distal gastric cancer, further deteriorating the clinical course [65]. Surgical gastroenterostomy (SGE) and self-expandable metallic stent (SEMS) placement have long been the main treatment options, depending on the patient’s overall status and prognosis; however, endoscopic ultrasound-guided gastroenterostomy (EUS-GE) has recently gathered significant attention as an effective therapeutic modality, warranting a concise description [66]. It involves the fusion of the stomach with the small intestine distally to the obstruction site, with or without guidewire assistance, utilizing a double-flanged lumen-apposing metallic stent (LAMS). The design of LAMS allows for improved anchoring and reduces the risk of migration compared to SEMS [67]. There are four main techniques to perform EUS-GE, namely, direct EUS-GE, the wireless endoscopic simplified technique (WEST), device-assisted EUS-GE, and EUS-guided double balloon-occluded gastrojejunostomy bypass (EPASS), with no significant differences in technical or clinical success rates among them [67,68,69]. On the contrary, EUS-GE is not feasible in cases of large-volume or malignant ascites, diffuse antral infiltration, or extensive peritoneal tumor implants [70]. A recent systematic review and meta-analysis of 36, mainly retrospective, studies (1846 patients with GOO due to malignant and/or benign etiologies) reported pooled technical and clinical success rates of 97% (95% CI, 95.9–98.0; I^2^ = 29.3%) and 90.6% (95% CI, 88.4–92.7; I^2^ = 60.9%), respectively [71]. The main advantage of EUS-GE over SEMS placement is its more durable effect [67]. Moreover, in a recent meta-analysis of 16 studies (1541 patients with malignant GOO), EUS-GE had a higher clinical success rate compared to SEMS (OR: 5.08; 95% CI, 3.42–7.55), with an equivalent safety profile (OR, 0.57; 95% CI, 0.29–1.14) [71]. In expert hands, several reports have also shown that EUS-GE demonstrates comparable clinical efficacy to SGE [67]. In fact, EUS-GE provides sustained symptomatic improvement comparable to SGE, with a lower migration rate compared to SEMS. These advantages have led to current guidelines recommending its use for the treatment of malignant GOO, provided that adequate expertise is available [70]. Regarding safety, a recent meta-analysis reported a pooled adverse event rate of 13% (95% CI, 10.3–15.7; I^2^ = 69.7%), including stent misdeployment, pneumoperitoneum, abdominal pain, bleeding, or gastric leak [72]. Stent misdeployment is considered the most serious complication of EUS-GE and a key factor limiting its routine application, occurring in approximately 10% of cases [73]. It results from failed deployment of either the proximal or distal flange of LAMS into the peritoneum or colon, potentially leading to significant adverse events. Therefore, to reduce the incidence of this feared complication, more echoendoscopists need to be trained in this procedure.

## 9. Advancements in EUS Staging of Gastric Cancer

EUS-elastography (EUS-E) and contrast-enhanced EUS (CE-EUS) are valuable add-on diagnostic tools currently used in various EUS applications, including lymph node assessment. In EUS-E, tissue stiffness is assessed and then displayed on a continuous color scale [74]. Given that malignant lymph nodes are generally stiffer than benign ones and the surrounding tissues, they are represented by different colors, allowing for accurate identification. A meta-analysis of seven studies found that the pooled sensitivity and specificity of EUS-E in discriminating between malignant and benign lymph nodes were 88% (95% CI, 83–92%) and 85% (95% CI, 79–89%), respectively [75]. Similarly, by applying contrast agents, CE-EUS enables the assessment of microvascularization, thereby providing valuable information about the nature of lesions. Typically, malignant lymph nodes demonstrate peripheral enhancement with loss of the normal capillary bed. Lisotti et al. [76], in their meta-analysis of four studies, reported a pooled sensitivity of 82.1% (75.1–87.7%) and a pooled specificity of 90.7% (85.9–94.3%) for CE-EUS in differentiating malignant from benign lymph nodes. Considering that the performance of conventional EUS in N staging is often suboptimal, both EUS-E and CE-EUS may serve as complementary tools in cases where EUS findings are inconclusive, either to support the diagnosis or to guide more invasive procedures such as EUS-FNA. The role of AI in the evaluation of gastric cancer has recently attracted considerable attention. While several studies have demonstrated the value of AI in improving the diagnostic accuracy of both conventional and image-enhanced endoscopy for diagnosing gastric cancer, only one report has addressed its implementation as an adjunctive tool to EUS. Uema et al. [77] introduced a two-step diagnostic model in which the tumor and gastric wall layers are first identified, followed by an assessment of invasion depth. This model was tested on both retrospectively and prospectively collected images and demonstrated diagnostic accuracy comparable to that of expert endoscopists (*p* = 0.95). Moreover, its application significantly improved diagnostic yield in cases with uncertain submucosal invasion based on conventional endoscopy.

## 10. Critical Appraisal of the Evidence

The current literature reports inconsistent outcomes regarding the performance of EUS in the T staging of gastric cancer, particularly in assessing the depth of invasion in EGC. However, it is crucial to acknowledge the gaps in existing research before drawing more reliable conclusions regarding the role of EUS in this context. First, there is significant heterogeneity in the reported outcomes across studies, which may partly arise from the different types of EUS used, as miniprobe EUS appears to be more accurate than conventional EUS in EGC [20,31,40]. Nevertheless, the miniprobe EUS is not routinely employed, and its use requires expertise that is not universally established. It is well recognized that the depth of submucosal invasion plays a key role in guiding the choice between endoscopic and surgical resection; however, there is currently no consensus on the definition of deep submucosal invasion by EUS. Some studies define it based on the irregularity of the submucosal layer, while others use specific thresholds, such as an invasion of 1 mm. To make things even more complicated, a few studies have proposed specific EUS patterns to define deep submucosal invasion and differentiate it from the frequently associated fibrosis [22,78]. Regarding N staging, the literature highlights disadvantages of EUS in both diagnosing and excluding lymph node metastases, which could in part be attributed to the significant variability among studies, especially in the criteria for malignancy, and the availability of EUS-FNA technology over time [45,46]. Given that EUS performance is highly operator-dependent, the lack of standardized diagnostic criteria for both T and N staging further contributes to inconsistencies in results. Another area of debate involves the use of EUS following neoadjuvant chemotherapy. EUS is not currently recommended for assessing response, as various studies have demonstrated limited accuracy for both T and N restaging, possibly due to inflammatory changes [58]. Nevertheless, other reports suggest that EUS may have prognostic value, as they found a significant correlation between tumor downstaging and patient prognosis [56,57,64]. Further studies are required to confirm the predictive role of EUS and to clarify whether tumor downstaging alone is sufficient to predict prognosis or if specific response criteria need to be established.

## 11. Conclusions

Accurate staging of gastric cancer is pivotal for appropriate patient management, especially in the early stages, to determine whether the patient is a candidate for endoscopic treatment. The effective application of any diagnostic modality, especially in such critical clinical contexts, necessitates a thorough awareness of its shortcomings as well as a clear understanding of the settings in which it is most effective. As previously discussed, EUS demonstrates high accuracy in advanced disease, whereas its performance in assessing invasion depth in EGC can be variable. This is due to the fact that various tumor-related factors, such as size, location, and ulceration, can affect its accuracy, leading to both overestimation and underestimation. It is essential that the echoendoscopist understands the influence of each specific factor to facilitate effective interpretation of the findings. Although not widely available, miniprobe EUS appears to be more suitable for early EGC due to its more detailed assessment of the gastric wall, which results in better performance in small tumors compared to conventional EUS. Additionally, the ability to perform endoscopic evaluation during the same session allows for a combined approach to be readily applied. Indeed, with advances in enhanced-imaging modalities (NBI and magnification), the accuracy of endoscopy in estimating invasion depth has improved [36]. Thus, when evaluating a patient with early gastric cancer, it may be reasonable to apply EUS, ideally using miniprobes, as a complementary tool when the endoscopic assessment of invasion depth is inconclusive and a decision regarding ESD indication is required. This strategy may also contribute to reduced costs and more efficient use of healthcare resources. We suggest that an integrated strategy with other imaging modalities, such as CT and PET–CT, also be considered for N staging, given the limited reliability of EUS in this context. Furthermore, one should bear in mind that even when inaccurate staging with EUS leads to non-curative endoscopic therapy, this does not preclude subsequent salvage treatment with radical gastrectomy to achieve a complete resection. A large meta-analysis including 4870 patients reported a favorable prognosis, in terms of both overall survival and disease-free survival, for patients who underwent salvage gastrectomy following non-curative endoscopic treatment [79]. In fact, patients with locally advanced disease, even in the presence of peritoneal carcinomatosis, may still benefit from advanced surgical approaches [80]. Notably, our review underscores a novel and increasingly recognized application of EUS, extending beyond its traditional diagnostic and staging roles. EUS may also serve a therapeutic role in patients with advanced, unresectable tumors causing GOO, as EUS-GE appears effective in symptom relief. Nonetheless, it is currently performed only in highly experienced centers and requires further standardization, as well as the training of echoendoscopists, before it can be broadly adopted. Similarly, despite promising results, the role of EUS-E and CE-EUS, particularly in the characterization of lymph node status, warrants further investigation. Finally, the role of AI as an adjunctive tool for EUS-guided gastric cancer staging also needs to be critically assessed.

## Figures and Tables

**Figure 1 diseases-13-00234-f001:**
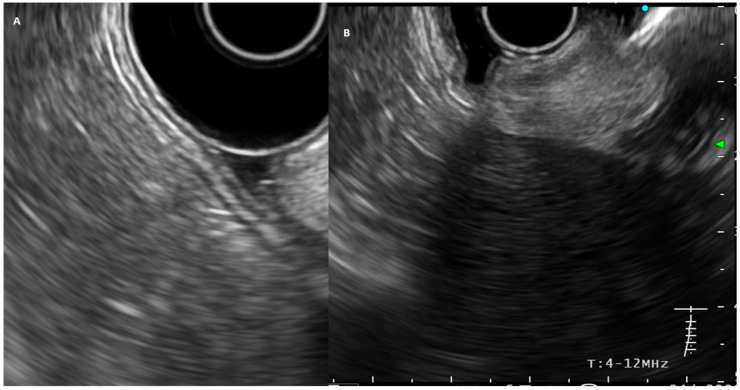
(**A**) EUS-image of the normal gastric wall. (**B**) EUS image of a gastric cancer with an echo-poor tumor involving the full thickness of the gastric wall (stage T3).

**Table 1 diseases-13-00234-t001:** Meta-analyses evaluating endoscopic ultrasound (EUS) performance for T staging in gastric cancer. NA: not available, T1M: mucosal invasion, T1SM: submucosal invasion, and T1SM1: minimal submucosal invasion.

Author	Year	Number of Studies	Stage Assessed	Sensitivity (%)	Specificity (%)	Accuracy (%)
Puli et al. [15]	2008	22	T1 T2 T3 T4	88.1 82.3 89.7 99.2	100 95.6 94.7 96.7	NA
Cardoso et al. [14]	2012	22	T (all stages) T1 T2 T3 T4	NA	NA	75 77 65 85 79
Mocelin et al. [18]	2015	66	T1–T2 vs. T3–T4 T1 vs. T2 T1M vs. T1SM	86 85 87	90 90 75	NA
Pei et al. [19]	2015	16	T1M T1SM T1M–T1SM1	76 62 90	72 78 67	NA
Shi et al. [20]	2019	17	Invasion depth in early gastric cancer (EGC)	87	67	NA

**Table 2 diseases-13-00234-t002:** Summary of findings on EUS assessment for tumor restaging after neoadjuvant chemotherapy.

Author and Year	Study Design	Number of Patients	Findings
Guo et al. 2014 [58]	Prospective	48	Overall accuracy for T restaging: 63%. Sensitivity and specificity for N restaging: 56% and 50%, respectively.
Bohle et al. 2017 [57]	Prospective	67	Correct T restaging: 22%. Overstaging rate: 60%. Sensitivity, specificity, and accuracy for N+ restaging: 53%, 54%, and 53%, respectively. Post-neoadjuvant T0–T2 stage, ≥2-step downstaging, and tumor thickness < 15 mm associated with longer recurrence-free survival.
Hoibian et al. 2021 [56]	Retrospective	97	Accuracy for T restaging: 44%; accuracy for N restaging: 49%. Overall survival significantly correlated with EUS response to neoadjuvant therapy (*p* = 0.01).
Sacerdotianu et al. 2022 [55]	Meta-Analysis	246	Pooled EUS sensitivity and specificity for restaging were as follows: 29% and 89% for T2, 71% and 49% for T3, 56% and 87% for T4, 45% and 86% for T1 + T2, 53% and 72% for N staging.

**Table 3 diseases-13-00234-t003:** Comparative evaluation of endoscopic ultrasound (EUS) and computed tomography (CT) in gastric cancer staging.

Modality	Comparative Advantages Over the Other
EUS	More accurate for early gastric cancer (EGC) staging More reliable for detecting and excluding lymph node involvement No radiation exposure
CT	Preferred modality for detecting distant metastases Greater accuracy in advanced lymph node involvement Fast, widely available, and non-invasive Staging accuracy not affected by ulceration

## Data Availability

No new data were created.

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
