# Peer review of "Endoscopic Ultrasound (EUS) in Gastric Cancer: Current Applications and Future Perspectives"

_diseases, 2025, doi:10.3390/diseases13080234_

Round 1

Reviewer 1 Report

Comments and Suggestions for Authors

A good summary of the current status of EUS in the staging of gastric cancer however there are several improvements that need to be included in the manuscrfipt

  1. The authors differentiate between T1 and T2 tumours but do not discuss at all the clinical need and importance of differentiating between Tis, T1a and T1b. The authors need to discuss the implications for lymph node involvement in these three types of depths, ie negligible for Tis, 2-5% for T1a and up to 30% for T1b. They need to discuss whether eus can determine and differentiate between these tumours based on layers or whether as per some recent studies can be based on the actual thickness. They also need to then furthe rdiscuss the implications of this for treatment as to whether the patients may be considered for endoscopic resection or radical surgical resection. Although this discussion occurs for differentiation of T1 v T2 this subset discussion is of great importnace
  2. The authors alos need to discuss salvage where the staging is incorrect, ie where it has overstaged. If the patient is indeed forund to have a poorer prognosis T1b with lymphatic invasion, or was found to have a T2 on histological resection or where the margins are incomplete, then there should be some discussion that this does then not preclude the patient then moving onto the standard treatment of a redical gastrectomy. This was not emphasised and would improve the manuscript.

Author Response

Dear Reviewer,

Thank you very much for taking the time to review this manuscript. Please find the detailed responses below and the corresponding revisions/corrections highlighted in the re-submitted files.

Comment 1: The authors differentiate between T1 and T2 tumours but do not discuss at all the clinical need and importance of differentiating between Tis, T1a and T1b.

Response: Thank you for pointing this out. We fully agree that the distinction between mucosal and submucosal cancer is of great significance and is perhaps the most critical factor when evaluating early-stage tumors. Indeed, we had not made this point clear in the original version of the manuscript. Therefore, we have now clarified its importance and have related it directly to the clinical decision-making process, regarding the selection of patients who are suitable candidates for endoscopic therapy.

“Once the diagnosis of early EGC is established, it is equally crucial to determine whether the tumor is confined to the mucosa or has invaded the submucosa, as well as to estimate the depth of invasion, in order to evaluate whether the patient is a candidate for endoscopic therapy.” Section: 4.2, Page: 5, Line: 144-147.

Comment 2: The authors need to discuss the implications for lymph node involvement in these three types of depths, ie negligible for Tis, 2-5% for T1a and up to 30% for T1b.

Response 2: Thank you for bringing this important issue to our attention. We have now added a brief discussion on this topic, highlighting both the differing incidence of nodal metastasis between T1a and T1b tumors and its clinical implications. Specifically, T1b tumors carry a considerable risk of nodal metastasis, which typically necessitates combined treatment with chemotherapy and surgery, similar to more advanced T stages. Therefore, we have emphasized the importance of thorough N staging in cases of T1b diagnosis before proceeding to surgery.

Moreover, there is a considerable difference in the incidence of lymph node metastasis between mucosal and submucosal tumors, with rates of approximately 2–5% for the former and up to 30% for the latter. This emphasizes the need for a thorough evaluation for nodal metastasis in cases of submucosal invasion, as these patients require a combined treatment approach of neoadjuvant chemotherapy and surgery.” Section 4.2, Page: 5, Line: 151-157

Comment 3: They need to discuss whether eus can determine and differentiate between these tumours based on layers or whether as per some recent studies can be based on the actual thickness. They also need to then furthe rdiscuss the implications of this for treatment as to whether the patients may be considered for endoscopic resection or radical surgical resection. Although this discussion occurs for differentiation of T1 v T2 this subset discussion is of great importance.

Response 3: Thank you for this insightful comment. This is indeed a very important issue, as it may contribute to the heterogeneity of the reported results due to variations in the definition of submucosal invasion, particularly in defining deep invasion that requires surgery. We had already mentioned this in the “Critical Appraisal” section (Page:12, Line: 402-408): “It is well recognized that the depth of submucosal invasion plays a key role in guiding the choice between endoscopic and surgical resection; however, there is currently no consensus on the definition of deep submucosal invasion by EUS. Some studies define it based on irregularity of the submucosal layer, while others use specific thresholds, such as an invasion of 1 mm. To make things even more complicated, a few studies have proposed specific EUS patterns to define deep submucosal invasion and differentiate it from the frequently associated fibrosis.

To further emphasize the significance of this aspect, we have added in the 4.2 section (Page 5, Line: 156-157), a description of how EUS discriminates between T1a and T1b tumors based on the layers : “On EUS, mucosal cancers are defined as lesions confined to the first and second wall layers, whereas submucosal cancers extend into the third layer”.

Additionally, we have noted that some studies used a specific threshold of 1 mm to define deep submucosal cancer and reported promising results regarding diagnostic accuracy: "To improve accuracy in estimating deep submucosal invasion and assessing the feasibility of endoscopic therapy, some studies have used a threshold of up to 1 mm, reporting promising results". Section 4.2, Page 6, Line: 171-174.

Comment 4: The authors alos need to discuss salvage where the staging is incorrect, ie where it has overstaged. If the patient is indeed forund to have a poorer prognosis T1b with lymphatic invasion, or was found to have a T2 on histological resection or where the margins are incomplete, then there should be some discussion that this does then not preclude the patient then moving onto the standard treatment of a redical gastrectomy. This was not emphasised and would improve the manuscript.

Response 4: We fully agree that this is of paramount importance, as an inaccurate initial staging may lead to non-curative endoscopic treatment. Therefore, we have clarified that an initial unsuccessful treatment decision does not preclude the patient from receiving definitive treatment with radical gastrectomy. To emphasize this point, we have added real-world data from a large meta-analysis, which reported favorable prognosis for patients who underwent surgery following non-curative endoscopic therapy compared to those who were only placed under surveillance.

“Furthermore, one should bear in mind that even when inaccurate staging with EUS leads to non-curative endoscopic therapy, this does not preclude subsequent salvage treatment with radical gastrectomy to achieve a complete resection. A large meta-analysis including 4,870 patients reported a favorable prognosis, in terms of both overall survival and disease-free survival, for patients who underwent salvage gastrectomy following non-curative endoscopic treatment”. Section 11, Page 13, Line: 445-450.

We hope we have thoroughly addressed all the valuable points you raised. Thank you once again for your careful review and insightful comments, which have helped improve our manuscript.

Reviewer 2 Report

Comments and Suggestions for Authors

This manuscript provides a comprehensive overview of the role of EUS in the diagnosis and treatment of gastric cancer, effectively reflecting the latest advancements while also offering practical clinical applications. I believe it holds significant academic value.

Author Response

Comment: This manuscript provides a comprehensive overview of the role of EUS in the diagnosis and treatment of gastric cancer, effectively reflecting the latest advancements while also offering practical clinical applications. I believe it holds significant academic value.

Response: We sincerely thank the reviewer for their positive and encouraging feedback. We are pleased that the reviewer recognizes the value of our work and its contribution to the field.

Reviewer 3 Report

Comments and Suggestions for Authors

Excellent review on a cutting edge topic. I have only some minor comments:

1) Maybe the authors could add a table on the studies on restaging after therapy

2) The authors should comment on the impact of mucosal incision (derived from ESD) techniques for tissue sampling of these lesions and their risk of adverse events (cite the relevant NMA: PMID: 31401022)

3) THe authors should comment on the potential prognostic impact of a correct staging and early diagnosis in this setting (cite the relevant paper: PMID: 34001385)

4) If the authors conducted a systematic review, they should report the literature search string

Author Response

Dear Reviewer,

Thank you very much for taking the time to review this manuscript. Please find the detailed responses below and the corresponding revisions/corrections highlighted in the re-submitted files.

Comment 1: Maybe the authors could add a table on the studies on restaging after therapy.

Response: Thank you for pointing this out. We have now added a table summarizing the findings of major studies evaluating the role of EUS in the restaging of gastric cancer. Page 8-9, Lines: 292-295

Comment 2: The authors should comment on the impact of mucosal incision (derived from ESD) techniques for tissue sampling of these lesions and their risk of adverse events (cite the relevant NMA: PMID: 31401022).

Response: Unfortunately, we were unable to identify the aforementioned citation. However, we sincerely thank you for your guidance.

Comment 3: THe authors should comment on the potential prognostic impact of a correct staging and early diagnosis in this setting (cite the relevant paper: PMID: 34001385).

Response: This is a very interesting aspect that has enhanced the quality of our manuscript. Indeed, it is important to emphasize treatment planning in cases of locally advanced tumors. This becomes particularly significant when inaccurate EUS staging occurs and a patient is mistakenly treated as having early-stage gastric cancer. You can find the relevant text in Section 11. Page 13, Lines: 450-452 : "In fact, patients with locally advanced disease, even in the presence of peritoneal carcinomatosis, may still benefit from advanced surgical approaches".

Comment 4: If the authors conducted a systematic review, they should report the literature search string.

Response: In fact, we did not conduct a systematic review. We simply described our search method to provide a clearer understanding of our work.

We hope we have thoroughly addressed all the valuable points you raised. Thank you once again for your careful review and insightful comments, which have helped improve our manuscript.

Reviewer 4 Report

Comments and Suggestions for Authors

This is an interesting paper. Please introduce any papers that discuss the difference between radial and linear scopes in diagnosing the depth of gastric cancer invasion in EUS diagnosis. In recent years, diagnosis has also been performed using magnifying endoscopy. Please also introduce papers that compare magnifying endoscopy and EUS in diagnosing gastric cancer.

Author Response

Dear Reviewer,

Thank you very much for taking the time to review this manuscript. Please find the detailed responses below and the corresponding revisions/corrections highlighted in the re-submitted files.

Comment 1: Please introduce any papers that discuss the difference between radial and linear scopes in diagnosing the depth of gastric cancer invasion in EUS diagnosis. 

Response: Thank you for pointing this out. We were already aware of the unique study comparing linear and radial echoendoscopes in T staging. Following your recommendation, we have now included this study in our review.

To our knowledge, only one study has directly compared linear and radial echoendoscopes for depth evaluation in EGC. This was a prospective study of 72 patients conducted by Lan et al., which found that linear EUS had significantly greater accuracy than radial EUS in diagnosing submucosal invasion (90.9% vs. 69.2%, P = 0.024). A potential advantage of linear EUS is the more feasible examination of tumors located in anatomically challenging regions of the stomach, such as the fundus and antrum, compared to radial EUS.Page: 6, Section: 4.2, Lines: 178-184.

Comment 2:  In recent years, diagnosis has also been performed using magnifying endoscopy. Please also introduce papers that compare magnifying endoscopy and EUS in diagnosing gastric cancer.

Response: Indeed, advancements in endoscopy have significantly improved its performance in the diagnosis and staging of gastric cancer. We had already mentioned this in the manuscript, along with the value of a combined approach using both endoscopy and EUS for staging gastric cancer. We conducted a thorough literature search for studies comparing magnifying endoscopy with EUS in the diagnosis and staging of gastric cancer; however, we did not identify any relevant studies suitable for inclusion in our review. Specifically, a recent study comparing WLE with EUS and magnifying endoscopy with NBI was excluded because it was published in Chinese (PMID: 35048617). Furthermore, we identified a meta-analysis also from China (Tittle: “A comparison of White light imaging (WLI), EUS and  ME-NBI for assessing the invasive depth of  early gastric cancer: a meta-analysis”) that assessed WLE, EUS, and ME-NBI for staging gastric cancer. However, a direct comparison between ME-NBI and EUS could not be made due to insufficient data provided in the study. Finally, preliminary data from a prospective study comparing ME-NBI with EUS (DOI: 10.1055/s-0044-1782720) were recently presented at “ESGE Days 2024.” However, we have chosen to include only data from fully published studies in our review.

We hope you understand this situation, and we would like to thank you again for your valuable recommendations.

Round 2

Reviewer 1 Report

Comments and Suggestions for Authors

Thank you for the revised manuscript. It addresses all the issues i raised and is now suitable for publication

Reviewer 3 Report

Comments and Suggestions for Authors

The manuscript is OK. Thank you!

Reviewer 4 Report

Comments and Suggestions for Authors

No additional comments.